# Household Income Is Related to Dietary Fiber Intake and Dietary Acid Load in People with Type 2 Diabetes: A Cross-Sectional Study

**DOI:** 10.3390/nu14153229

**Published:** 2022-08-07

**Authors:** Fuyuko Takahashi, Yoshitaka Hashimoto, Yukiko Kobayashi, Ayumi Kaji, Ryosuke Sakai, Takuro Okamura, Naoko Nakanishi, Saori Majima, Hiroshi Okada, Takafumi Senmaru, Emi Ushigome, Mai Asano, Masahide Hamaguchi, Masahiro Yamazaki, Wataru Aoi, Masashi Kuwahata, Michiaki Fukui

**Affiliations:** 1Department of Endocrinology and Metabolism, Graduate School of Medical Science, Kyoto Prefectural University of Medicine, Kyoto 602-8566, Japan; 2Department of Diabetes and Endocrinology, Matsushita Memorial Hospital, Moriguchi 570-8540, Japan; 3Department of Nutrition Science, Graduate School of Life and Environmental Sciences, Kyoto Prefectural University, Kyoto 606-8522, Japan

**Keywords:** household income, nutrition, diet, diet quality, type 2 diabetes mellitus

## Abstract

Household income was related to habitual dietary intake in general Japanese people. This cross-sectional study investigated the relationship between household income and habitual dietary intake in people with type 2 diabetes mellitus (T2DM). Household income was evaluated using a self-reported questionnaire and categorized into high and low household income. Nutritional status was assessed using a brief-type self-administered diet history questionnaire. Among 128 men and 73 women, the proportions of participants with low household income were 67.2% (*n* = 86/128) in men and 83.6% (*n* = 61/73) in women. Dietary fiber intake (11.3 ± 4.2 vs. 13.8 ± 6.0 g/day, *p* = 0.006) was lower, and dietary acid load, net endogenous aid production score (NEAP) (51.7 ± 10.5 vs. 46.8 ± 10.4 mEq/day, *p* = 0.014) and potential renal acid load score (PRAL) (9.5 ± 10.7 vs. 3.7 ± 14.1 mEq/day, *p* = 0.011) were higher in men with low household income than in those without. Multivariable linear regression analyses demonstrated that log (dietary fiber intake) in men with low household income was lower than that in those with high household income after adjusting for covariates (2.35 [2.26–2.44] vs. 2.52 [2.41–2.62], *p* = 0.010). Furthermore, NEAP (54.6 [51.7–57.4] vs. 45.8 [42.5–49.2], *p* <0.001) in men with low household income were higher than in those with high household income after adjusting for covariates. Contrastingly, household income was not related to diet quality in women. This study showed that household income was related to dietary fiber intake and dietary acid load in men but not in women.

## 1. Introduction

All over the world, the population of people with type 2 diabetes mellitus (T2DM) continues to increase [1]. Socioeconomic status, which consists of educational level, occupation, living status, and household income, affects the prevalence of T2DM [2]. In particular, low household income has been related to the prevalence of T2DM [3,4]. Among people with T2DM, those with low income have been shown to have worse glycemic control than those with high income [5]. Moreover, low household income is found to be the risk of mortality in general populations [6]. Therefore, people with low household income are considered to have various risks.

According to data from the 2014 National Health and Nutrition Survey in Japan, a lower household income was related to higher carbohydrate intake and lower vegetable intake [7]. Moreover, a previous study revealed the association between low household income and low dietary fiber intake [8]. Among people with T2DM, dietary fiber intake has been shown to improve glycemic control, decrease hyperinsulinemia, and decrease plasma lipid concentrations [9]. Dietary fiber intake has reportedly been associated with all-cause mortality [10,11]. 

Moreover, dietary acid load has been revealed as a risk factor for metabolic syndrome [12], T2DM [13,14], hypertension [15], and mortality [16]. Dietary acid load score includes potential renal acid load (PRAL) and net endogenous acid production (NEAP). PRAL reflects the rates of intestinal absorption of contributing balances of nutrient ions for protein, potassium, calcium, and magnesium, as well as the dissociation of phosphate at pH 7.4 [17]. NEAP, estimated by the ratio of protein to potassium content in a diet, mirrors acid balance and is known as the risk of the chronic kidney disease advancement [18]. 

However, the relationship between household income and habitual dietary intake, especially dietary fiber intake and dietary acid load, in people with T2DM is unclear; thus, this cross-sectional study proposed to examine this association. 

## 2. Method

### 2.1. Study Design, Setting and Participants

This cross-sectional study was included in the prospective KAMOGAWA-DM cohort study, running since 2014 [19]. This cohort study involved outpatients from the Department of Endocrinology and Metabolism, Kyoto Prefectural University of Medicine Hospital (Kyoto, Japan). The goal of this cohort study is to reveal the natural history of people with diabetes. The patients were invited to participate by their primary doctors, and those who agreed were included in this cohort study. All participants provided written informed consent. The present study was carried out in accordance with the Declaration of Helsinki with the approval of the Local Research Ethics Committee (No. RBMR-E-466-6). The inclusion criterion was the capability of responding to the questionnaires, including the brief-type self-administered diet history questionnaire (BDHQ), from January 2016 to February 2021. The exclusion criteria were non-T2DM; extremely low or high energy intake (<600 or >4000 kcal/day), as extremely low or high energy intake is unnatural [20]; incomplete questionnaire; and unknown household income.

### 2.2. Questionnaire Regarding Lifestyle Characteristics and Household Income

Participants were given a standardized questionnaire to assess lifestyle factors and household income. According to the answer to the questionnaire, participants were categorized as non-smokers and current smokers. Additionally, participants were categorized as non-exercisers and exercisers based on their performance, or lack thereof, of any type of sport at least one time per week. Educational level was evaluated with the following response options: “elementary school”, “junior high school”, “high school”, “technical college”, “vocational school”, “college”, and “graduate school”, and educational background of “elementary school” or “junior high school” was defined as <12 years [21]. Household income was evaluated with the following response options: “<3,000,000 JPY”, “3,000,000–5,000,000 JPY”, “5,000,000–8,000,000 JPY”, “≥8,000,000 JPY”, and “unknown or declined to answer” [22]. The average salary at that time of this study was JPY 4,360,000 [23]. Therefore, household income of “<3,000,000 JPY” or “3,000,000–5,000,000 JPY” was defined as low household income, whereas that of “5,000,000–8,000,000 JPY” or “≥8,000,000 JPY” was defined as high household income in this study [23]. 

### 2.3. Participant Data

Body mass index (BMI) was obtained as follows: body weight (kg) divided by height squared (m^2^). Ideal body weight (IBW) was determined as follows: IBW (kg) = 22 × (height [m])^2^ [24].

Fasting plasma glucose, glycosylated hemoglobin (HbA1c), uric acid, creatinine, triglycerides, and high-density lipoprotein cholesterol concentrations were analyzed using venous blood samples from all participants after a night of fasting. The estimated glomerular filtration rate (eGFR [mL/min/1.73 m^2^]) was estimated using the Japanese Society of Nephrology equation [25]. Renal failure was defined as eGFR <30 mL/min per 1.73 m^2^ [26]. Blood pressure was tested with an HEM-906 device (OMRON, Kyoto, Japan). Additionally, data on the use of medications, including insulin and antihypertensives, were gathered from the patients’ medical records. Hypertension was defined as systolic blood pressure of ≥140 mmHg and/or diastolic blood pressure of ≥90 mmHg, and/or use of antihypertensive drugs.

### 2.4. Estimation and Assessment of Habitual Food and Nutrient Intake

To assess habitual food and nutrient intake, the BDHQ, a dietary recall tool that estimates a respondent’s dietary intake of 58 items over the past month, was utilized [20]. The details and validity of BDHQ have been presented previously [27]. Data on energy (kcal/day); protein (g/day), including animal and vegetable proteins; fat (g/day); carbohydrate (g/day); fiber (g/day); phosphorus (mg/day); potassium (mg/day); magnesium (mg/day); calcium (mg/day); and alcohol (g/day) intakes were obtained from the BDHQ. Energy (kcal/IBW/day), fat (g/IBW/day), carbohydrate (g/IBW/day), total protein (g/IBW/day), animal protein (g/IBW/day), and vegetable protein (g/IBW/day) intakes were obtained. The carbohydrate to fiber intake ratio was calculated as follows: carbohydrate intake divided by fiber intake [28]. Alcohol consumption was also obtained, and habitual alcohol consumption was determined as that >20 g/day [29].

PRAL and NEAP were estimated as the following equations: PRAL (mEq/day) = 0.037 × phosphorus (mg/day) + 0.49 × protein (g/day) − 0.026 × magnesium (mg/day) − 0.021 × potassium (mg/day) − 0.013 × calcium (mg/day) [30] and NEAP (mEq/day) = −10.2 + (54.5 × protein [g/day]/potassium [mEq/day]) [31].

### 2.5. Statistical Analysis

Data are presented as means ± standard deviations or frequencies of potential confounding variables. The chi-square test was used for categorical variables, and the Student’s *t*-test was used for continuous variables to assess the statistical significance of differences between groups. Moreover, because the characteristics and dietary intakes differed between men and women, the data were analyzed by sex.

NEAP was equal variance. Although dietary fiber intake was not equal variance, logarithmic dietary fiber intake was equal variance. Therefore, NEAP and log (dietary fiber intake) were used for multivariable linear regression to assess the association between household income and log (dietary fiber intake) and dietary acid load. Multivariable linear regression analyses were executed, and geometric means with 95% confidence intervals were calculated, after adjusting for age, sex, BMI, the duration of diabetes, exercise habit, smoking habit, HbA1c, triglycerides, presence of hypertension, energy intake and alcohol consumption. Age, duration of diabetes, BMI, HbA1c, triglycerides and presence of hypertension are known to effect diet [32,33,34,35]. Exercise, smoking and drinking alcohol affected glycemic control, which are associated with diet therapy, including dietary fiber intake [36,37,38]. Increased energy intake results in a relatively high dietary fiber intake.

Statistical analyses were conducted using JMP software (version 13.2; SAS Institute Inc., Cary, NC, USA) and EZR (Saitama Medical Center, Jichi Medical University, Saitama, Japan) [39]. Differences with *p* values < 0.05 were considered statistically significant.

## 3. Results

In total, 338 people were contained in this study. We excluded 137 people: 24 without T2DM, 3 with hyper- or hypo-nutrition, 84 who failed to complete the questionnaire and 26 whose household income was unknown; thus, the final research population comprised 201 people (128 men and 73 women; Figure 1).

The clinical characteristics of study participants are sum up in Table 1. Mean age and BMI were 68.3 ± 9.5 years and 23.9 ± 3.3 kg/m^2^ in men and 70.4 ± 7.2 years and 23.5 ± 3.9 kg/m^2^ in women, respectively. The percentage of participants with high household income were 32.8% (*n* = 42/128) and 16.4% (*n* = 12/73) in men and women, respectively. Mean dietary fiber intake was 12.1 ± 5.0 g/day in men and 12.3 ± 4.9 g/day in women. Mean PRAL and NEAP were 7.6 ± 12.2 mEq/day and 50.1 ± 10.7 mEq/day in men and 3.7 ± 13.1 mEq/day and 47.0 ± 10.6 mEq/day in women, respectively.

Table 2 presents the results of clinical characteristics according to household income. People with low household intake were older than those with high household intake (70.4 ± 7.7 vs. 65.3 ± 10.4 years, *p* < 0.001). The percentage of men in people with low household intake was lower than that with high household intake (58.5 vs. 77.8%, *p* = 0.019). Dietary fiber intake in people with low household income was lower than that in those with high household income (11.7 ± 4.5 vs. 13.5 ± 5.9 g/day, *p* = 0.028). Dietary fiber intake in men with low household income was lower than that in those with high household income (11.3 ± 4.2 vs. 13.8 ± 6.0 g/day, *p* = 0.006). PRAL (9.5 ± 10.7 vs. 3.7 ± 14.1 mEq/day, *p* = 0.011) and NEAP (51.7 ± 10.5 vs. 46.8 ± 10.4 mEq/day, *p* = 0.014) in men with low household income were higher than in those with high household income.

Furthermore, we investigated the association of dietary fiber intake and NEAP with household income (Table 3). Log (dietary fiber intake) with low household intake tended to be lower than that with high household income (2.38 [2.30–2.46] vs. 2.47 [2.37–2.57], *p* = 0.088). Log (dietary fiber intake) in men with low household income was lower than that in those with high household income after adjusting for covariates (2.35 [2.26–2.44] vs. 2.52 [2.41–2.62], *p* = 0.010). Furthermore, NEAP (54.6 [51.7–57.4] vs. 45.8 [42.5–49.2], *p* < 0.001) in men with low household income were higher than in those with high household income after adjusting for covariates. In contrast, household income was not related to dietary fiber intake and dietary acid load in women after adjusting for covariates. 

The difference between included and excluded participants with T2DM was showed in Appendix A. HbA1c in included people was higher than that in excluded people (7.3 ± 0.9 vs. 7.0 ± 0.8 %, *p* = 0.032). Exercise habit were different between included and excluded participants with T2DM (57.7 vs. 38.9 %, *p* = 0.002). The other characteristics were not different between included and excluded participants with T2DM.

## 4. Discussion

This study verified the relationship between household income and habitual dietary intake, especially dietary fiber intake and dietary acid load, in people with T2DM. The results of this study demonstrated that household income was related to dietary fiber intake and dietary acid load in men but not in women.

In the present study, men with low household income consumed lower dietary fiber than those with high household income, and the presence of hypertension in men with low household income was more prevalent than that in those with high household income. Previous studies found that there was an association between household income and vegetable intake [8,40]. This might because that although people are aware that vegetables are good for their health, price of vegetables may be a barrier to purchase vegetables, especially for those with low household income [41]. There is a relationship between dietary fiber and glycemic control, insulin sensitivity and lipid concentration [9]. Dietary fiber intake is reportedly related to blood pressure [42]. Additionally, higher dietary fiber intake is reportedly associated with a lower risk of all-cause death [10,11,43]. Taking these finding together, adequate dietary fiber intake is recommended for people with T2DM; thus, we should pay attention to dietary fiber intake among men with low household income.

Furthermore, PRAL and NEAP in men with low household income were higher than in those with high household income in this study. PRAL and NEAP are parameters of dietary acid load and exhibit higher values in diets containing a lot of acidogenic foods, such as meat and fish, and a lack of alkaline foods, such as fruits and vegetables [44]. Previously, PRAL and NEAP were reported to have positive associations with blood pressure [45]. High PRAL is recognized as a risk of cardiovascular diseases [46], and high NEAP is known to be associated with hypertension [47]. Therefore, improving dietary quality, such as dietary fiber intake and dietary acid load, potentially decreases the presence of hypertension and cardiovascular disease in men with low household income.

Previously, a relationship between household income and glycemic control in people with T2DM has been found [5]. However, household income was not related to glycemic control in the current study. Participants in this study were limited to those who were continuously visiting diabetes outpatient clinics and receiving treatment; thus, there might not have been an association between glycemic control and household income.

In the present study, an association between household income and dietary fiber intake or dietary acid load was found in men but not in women. A previous study showed that women tended to practice dietary self-care behaviors more than men [48]. Moreover, women have tended to purchased vegetables and fruits because they regarded vegetables and fruits were healthy [41]. Taking these finding together, household income might not relate to dietary fiber intake and dietary acid load in women in the present study. Therefore, a higher interest in dietary treatment among women might have reduced the effect of household income on diet. 

The present study has certain limitations. First, socioeconomic status factors other than household income were not evaluated. Second, household income data were based on personal reporting, and thus the accuracy of the data was uncertain. Moreover, the number of participants, especially extreme incomes, were not enough. Therefore, we need further research with more participants and used the different cut-off. Third, since this study was a cross-sectional study, we could not confirm a causal relationship. Fourth, the validation of BDHQ has been showed previously [27]. However, the Pearson correlation coefficients between the dietary record and the BDHQ is around *r* = 0.60, which is a little low. Finally, all study participants were exclusively outpatients; therefore, the generalizability of the results to people with untreated T2DM is unclear.

## 5. Conclusions

This study showed that household income was related to dietary fiber intake and dietary acid load in men but not in women. Better dietary quality is important for people with T2DM; thus, clinicians and dieticians should pay attention to poor diet quality among men with low household income.

## Figures and Tables

**Figure 1 nutrients-14-03229-f001:**
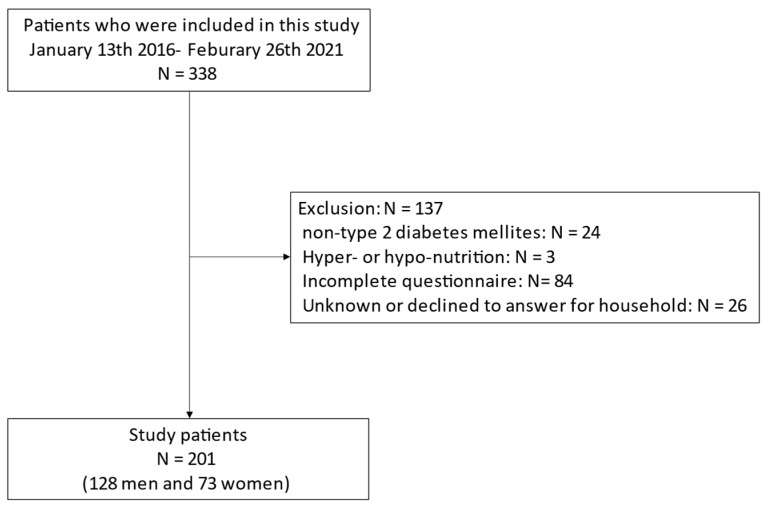
Inclusion and exclusion flow.

**Table 1 nutrients-14-03229-t001:** Clinical characteristics of study participants.

	All N = 201	Men N = 128	Women N = 73	*p*
Age (years)	69.0 (8.8)	68.3 (9.5)	70.4 (7.2)	0.097
Duration of diabetes (years)	17.7 (11.0)	17.4 (10.8)	18.2 (11.4)	0.651
Family history of diabetes (+)	40.8 (82)	32.8 (42)	54.8 (40)	0.004
Height (cm)	162.2 (9.3)	167.6 (6.2)	152.8 (5.6)	<0.001
Body weight (kg)	62.8 (12.2)	67.3 (11.0)	55.1 (10.1)	<0.001
Body mass index (kg/m^2^)	23.8 (3.5)	23.9 (3.3)	23.5 (3.9)	0.462
SBP (mmHg)	130.4 (16.2)	130.7 (16.2)	129.9 (16.4)	0.737
DBP (mmHg)	74.1 (11.5)	75.4 (11.8)	71.8 (10.7)	0.028
Antihypertensive drugs (+)	61.2 (123)	61.7 (79)	60.3 (44)	0.959
Presence of hypertension (+)	68.2 (137)	68.0 (87)	68.5 (50)	1.000
Insulin (+)	23.9 (48)	21.9 (28)	27.4 (20)	0.477
Smoking (+)	14.9 (30)	19.5 (25)	6.8 (5)	0.026
Habit of exercise (+)	57.7 (116)	54.7 (70)	63.0 (46)	0.317
Education level (<12 years)	12.4 (25) (no data 4.0 [8])	12.2 (15)	14.3 (10)	0.847
Married status (married/divorce/not married/bereavement)	74.6 (150)/ 11.0 (22)/ 6.5 (13)/ 4.5 (9) (no data 3.5 [7])	78.0 (96)/ 10.6 (13)/ 8.1 (10)/ 3.3 (4)	76.1 (54)/ 12.7 (9)/ 4.2 (3)/ 7.0 (5)	0.454
HbA1c (mmol/mol)	55.9 (9.9)	56.2 (10.7)	55.5 (8.5)	0.667
HbA1c (%)	7.3 (0.9)	7.3 (1.0)	7.2 (0.8)	0.667
Plasma glucose (mmol/L)	8.0 (2.1)	8.2 (2.3)	7.7 (1.7)	0.113
Creatinine (umol/L)	75.7 (36.4)	83.1 (39.1)	62.7 (26.8)	<0.001
eGFR (mL/min/1.73 m^2^)	69.7 (21.2)	69.7 (21.6)	69.6 (20.7)	0.973
Renal failure (+)	5.0 (10)	4.7 (6)	5.5 (4)	1.000
Uric acid (umol/L)	301.2 (90.0)	316.6 (93.5)	274.3 (77.0)	<0.001
Triglycerides (mmol/L)	1.5 (0.9)	1.6 (1.0)	1.4 (0.7)	0.103
HDL cholesterol (mmol/L)	1.5 (0.4)	1.5 (0.4)	1.7 (0.4)	<0.001
Household income (high)	26.9(54)	32.8 (42)	16.4 (12)	0.019
Total energy intake (kcal/day)	1727.7 (509.0)	1841.0 (494.6)	1529.1 (474.9)	<0.001
Energy intake (kcal/IBW kg/day)	29.8 (8.7)	29.9 (8.3)	29.8 (9.3)	0.924
Total protein intake (g/day)	72.8 (27.6)	74.7 (27.8)	69.4 (27.1)	0.190
Protein intake (g/IBW kg/day)	1.3 (0.5)	1.2 (0.5)	1.3 (0.5)	0.062
Protein intake (% Energy)	16.8 (3.3)	16.1 (3.3)	18.0 (3.0)	<0.001
Animal protein intake (g/day)	44.6 (22.3)	45.5 (22.5)	43.2 (22.0)	0.482
Animal protein intake (g/IBW kg/day)	0.8 (0.4)	0.7 (0.4)	0.8 (0.4)	0.081
Vegetable protein intake (g/day)	28.1 (8.7)	29.2 (8.7)	26.2 (8.3)	0.017
Vegetable protein intake (g/IBW kg/day)	0.5 (0.1)	0.5 (0.1)	0.5 (0.2)	0.123
Total fat intake (g/day)	55.7 (21.1)	57.7 (20.3)	52.3 (22.3)	0.082
Fat intake (g/IBW kg/day)	1.0 (0.4)	0.9 (0.3)	1.0 (0.4)	0.130
Fat intake (% Energy)	28.9 (6.5)	28.1 (6.4)	30.3 (6.4)	0.022
Total carbohydrate intake (g/day)	215.4 (68.0)	229.4 (69.3)	190.9 (58.6)	<0.001
Carbohydrate intake (g/IBW kg/day)	3.7 (1.1)	3.7 (1.2)	3.7 (1.1)	0.903
Carbohydrate intake (% Energy)	50.4 (8.8)	50.3 (9.3)	50.6 (8.1)	0.833
Dietary fiber intake (g/day)	12.2 (5.0)	12.1 (5.0)	12.3 (4.9)	0.785
Carbohydrate/fiber ratio	19.4 (7.1)	20.8 (7.6)	16.8 (5.5)	<0.001
Alcohol consumption (g/day)	7.8 (17.0)	11.8 (20.1)	0.6 (3.3)	<0.001
PRAL (mEq/day)	6.2 (12.6)	7.6 (12.2)	3.7 (13.1)	0.036
NEAP (mEq/day)	49.0 (10.7)	50.1 (10.7)	47.0 (10.6)	0.049

Data were expressed as mean (standard deviation) or percentage (number). The difference between group was evaluated by Student’s *t*-test or chi-square test. SBP, systolic blood pressure; DBP, diastolic blood pressure; eGFR, estimated glomerular filtration rate; HDL, high-density lipoprotein; IBW, ideal body weight; PRAL, potential renal acid load score; NEAP, net endogenous acid production score.

**Table 2 nutrients-14-03229-t002:** Clinical characteristics according to household income.

	All			Men			Women		
	Low N = 147	High N = 54	*p*	Low N = 86	High N = 42	*p*	Low N = 61	High N = 12	*p*
Age (years)	70.4 (7.7)	65.3 (10.4)	<0.001	70.4 (8.3)	63.9 (10.4)	<0.001	70.5 (6.8)	70.0 (9.4)	0.831
Sex (men)	58.5 (86)	77.8 (42)	0.019	-	-	-	-	-	-
Duration of diabetes (years)	19.2 (11.6)	13.6 (7.9)	0.001	19.4 (11.6)	13.4 (7.6)	0.003	18.9 (11.8)	14.3 (9.1)	0.197
Family history of diabetes (+)	42.2 (62)	37.0 (20)	0.620	34.9 (30)	28.6 (12)	0.607	52.5 (32)	66.7 (8)	0.557
Height (cm)	161.3 (9.5)	164.7 (8.3)	0.021	167.2 (6.8)	168.3 (4.8)	0.372	153.0 (5.7)	152.2 (5.3)	0.686
Body weight (kg)	61.5 (12.2)	66.4 (11.6)	0.012	66.0 (11.3)	69.9 (10.2)	0.062	55.2 (10.6)	54.2 (7.3)	0.748
Body mass index (kg/m^2^)	23.5 (3.6)	24.4 (3.3)	0.144	23.5 (3.2)	24.7 (3.4)	0.071	23.6 (4.1)	23.3 (2.5)	0.857
SBP (mmHg)	130.9 (17.0)	129.0 (13.8)	0.457	131.2 (17.6)	129.8 (13.0)	0.642	130.6 (16.4)	126.4 (16.6)	0.423
DBP (mmHg)	73.3 (11.6)	76.4 (11.0)	0.085	74.4 (11.9)	77.6 (11.3)	0.154	71.6 (11.0)	72.3 (9.3)	0.839
Antihypertensive drugs (+)	66.7 (98)	46.3 (25)	0.014	69.8 (60)	45.2 (19)	0.013	62.3 (38)	50.0 (6)	0.636
Presence of hypertension (+)	74.1 (109)	51.9 (28)	0.005	75.6 (65)	52.4 (22)	0.015	72.1 (44)	50.0 (6)	0.243
Insulin (+)	23.8 (35)	24.1 (13)	1.000	22.1 (19)	21.4 (9)	1.000	26.2 (16)	33.3 (4)	0.880
Smoking (+)	11.6 (17)	24.1 (13)	0.048	14.0 (12)	31.0 (13)	0.041	8.2 (5)	0.0 (0)	0.687
Habit of exercise (+)	59.2 (87)	53.7 (29)	0.592	57.0 (49)	50.0 (21)	0.579	62.3 (38)	66.7 (8)	1.000
Education level (<12 years)	14.9 (21)	7.7 (4)	0.280	14.5 (12)	7.5 (3)	0.418	15.5 (9)	8.3 (1)	0.846
Married status (married/divorce/not married/bereavement)	73.9 (105)/ 13.4 (19)/ 7.7 (11)/ 4.9 (7)	86.5 (45)/ 5.8 (3)/ 3.8 (2)/ 3.8 (2)	0.297	74.7 (62)/ 12.0 (10)/ 9.6 (8)/ 3.6 (3)	85.0 (34)/ 7.5 (3)/ 5.0 (2)/ 2.5 (1)	0.634	72.9 (43)/ 15.3 (9)/ 5.1 (3)/ 6.8 (4)	91.7 (11)/ 0 (0)/ 0 (0)/ 8.3 (1)	0.401
HbA1c (mmol/mol)	55.7 (9.7)	56.6 (10.5)	0.562	55.9 (10.5)	56.8 (11.0)	0.637	55.5 (8.6)	55.9 (8.8)	0.866
HbA1c (%)	7.2 (0.9)	7.3 (1.0)	0.562	7.3 (1.0)	7.3 (1.0)	0.637	7.2 (0.8)	7.3 (0.8)	0.866
Plasma glucose (mmol/L)	8.1 (2.3)	7.9 (1.8)	0.598	8.4 (2.6)	7.8 (1.8)	0.170	7.6 (1.7)	8.2 (1.9)	0.261
Creatinine (umol/L)	77.1 (39.1)	71.9 (27.9)	0.368	85.5 (43.4)	78.2 (28.2)	0.324	65.3 (28.4)	49.7 (9.7)	0.066
eGFR (mL/min/1.73 m^2^)	67.3 (20.7)	76.3 (21.3)	0.008	67.5 (21.1)	74.4 (22.1)	0.088	67.1 (20.5)	82.8 (17.5)	0.015
Renal failure (+)	5.4 (8)	3.7 (2)	0.891	4.7 (4)	4.8 (2)	1.000	6.6 (4)	0 (0)	0.827
Uric acid (mmol/L)	299.9 (91.6)	304.7 (86.3)	0.743	314.4 (101.3)	321.1 (76.0)	0.708	279.6 (71.8)	247.3 (98.8)	0.187
Triglycerides (mmol/L)	1.4 (0.8)	1.7 (0.9)	0.103	1.5 (1.0)	1.7 (0.9)	0.328	1.3 (0.6)	1.6 (1.1)	0.280
HDL cholesterol (mmol/L)	1.5 (0.5)	1.5 (0.4)	0.436	1.5 (0.4)	1.5 (0.4)	0.928	1.7 (0.4)	1.6 (0.5)	0.816
Total energy intake (kcal/day)	1679.9 (498.7)	1857.8 (518.9)	0.028	1782.8 (479.7)	1960.2 (508.9)	0.056	1534.9 (492.6)	1499.6 (389.7)	0.816
Energy intake (kcal/IBW kg/day)	29.4 (8.8)	31.1 (8.2)	0.217	29.1 (8.2)	31.5 (8.3)	0.116	29.8 (9.6)	29.5 (8.1)	0.922
Total protein intake (g/day)	71.3 (27.1)	76.7 (29.0)	0.223	72.1 (36.5)	80.1 (30.0)	0.129	70.3 (28.1)	64.9 (22.4)	0.536
Protein intake (g/IBW kg/day)	1.3 (0.5)	1.3 (0.5)	0.697	1.2 (0.5)	1.3 (0.5)	0.199	1.4 (0.6)	1.3 (0.4)	0.578
Protein intake (% Energy)	16.9 (3.4)	16.4 (3.1)	0.292	16.0 (3.4)	16.1 (3.1)	0.887	18.2 (3.0)	17.2 (3.2)	0.308
Animal protein intake (g/day)	44.2 (22.2)	45.9 (22.7)	0.619	44.3 (21.7)	47.9 (24.3)	0.405	43.9 (23.2)	39.2 (14.9)	0.499
Animal protein intake (g/IBW kg/day)	0.8 (0.4)	0.8 (0.4)	0.902	0.7 (0.4)	0.8 (0.4)	0.488	0.9 (0.5)	0.8 (0.3)	0.512
Vegetable protein intake (g/day)	27.2 (8.1)	30.8 (9.7)	0.009	27.8 (8.0)	32.2 (9.3)	0.006	26.3 (8.1)	25.7 (9.6)	0.819
Vegetable protein intake (g/IBW kg/day)	0.5 (0.1)	0.5 (0.2)	0.094	0.5 (0.1)	0.5 (0.2)	0.017	0.5 (0.1)	0.5 (0.2)	0.959
Total fat intake (g/day)	54.7 (21.4)	58.5 (20.2)	0.250	56.2 (20.0)	60.8 (20.9)	0.230	52.6 (23.4)	50.8 (16.2)	0.802
Fat intake (g/IBW kg/day)	1.0 (0.4)	1.0 (0.3)	0.718	0.9 (0.3)	1.0 (0.3)	0.317	1.0 (0.5)	1.0 (0.3)	0.849
Fat intake (% Energy)	29.1 (6.7)	28.4 (5.8)	0.521	28.3 (6.9)	27.8 (5.5)	0.663	30.2 (6.4)	30.8 (6.5)	0.798
Total carbohydrate intake (g/day)	208.3 (66.3)	234.9 (69.5)	0.014	220.7 (69.1)	247.2 (67.1)	0.042	190.7 (58.4)	191.8 (62.2)	0.955
Carbohydrate intake (g/IBW kg/day)	3.6 (1.1)	3.9 (1.1)	0.109	3.6 (1.2)	4.0 (1.1)	0.094	3.7 (1.1)	3.8 (1.3)	0.792
Carbohydrate intake (% Energy)	50.2 (9.1)	50.9 (8.2)	0.626	50.0 (9.7)	50.9 (8.4)	0.618	50.5 (8.1)	50.9 (8.1)	0.859
Dietary fiber intake (g/day)	11.7 (4.5)	13.5 (5.9)	0.028	11.3 (4.2)	13.8 (6.0)	0.006	12.4 (4.8)	12.1 (5.7)	0.876
Carbohydrate/fiber ratio	19.4 (7.2)	19.3 (7.1)	0.899	21.4 (7.7)	19.7 (7.3)	0.236	16.7 (5.3)	17.8 (6.5)	0.497
Alcohol consumption (g/day)	7.1 (17.1)	9.4 (16.7)	0.398	11.8 (21.1)	12.0 (18.2)	0.949	0.7 (3.6)	0.5 (0.8)	0.881
PRAL (mEq/day)	7.1 (12.4)	3.6 (13.1)	0.088	9.5 (10.7)	3.7 (14.1)	0.011	3.7 (13.8)	3.6 (9.2)	0.989
NEAP (mEq/day)	49.7 (10.9)	46.9 (10.1)	0.102	51.7 (10.5)	46.8 (10.4)	0.014	46.9 (11.0)	47.4 (9.2)	0.883

Data were expressed as mean (standard deviation) or percentage (number). The difference between group was evaluated by Student’s t-test or chi-square test. SBP, systolic blood pressure; DBP, diastolic blood pressure; eGFR, estimated glomerular filtration rate; HDL, high-density lipoprotein; IBW, ideal body weight; PRAL, potential renal acid load score; NEAP, net endogenous acid production score.

**Table 3 nutrients-14-03229-t003:** The adjusted correlation of dietary fiber intake or net endogenous acid production score with household income.

	All			Men			Women		
	Household Income (Low)	Household Income (High)	*p*	Household Income (Low)	Household Income (High)	*p*	Household Income (Low)	Household Income (High)	*p*
Model 1									
Log dietary fiber	2.38 (2.32–2.45)	2.54 (2.43–2.65)	0.014	2.33 (2.25–2.41)	2.59 (2.47–2.71)	<0.001	2.44 (2.34–2.55)	2.39 (2.15–2.62)	0.663
NEAP (mEq/day)	49.6 (47.9–51.4)	45.4 (42.4–48.4)	0.017	52.3 (50.0–54.5)	45.7(42.5–49.0)	0.002	46.9 (44.2–49.7)	47.5 (41.2–53.7)	0.878
Model 2									
Log dietary fiber	2.37 (2.27–2.47)	2.51 (2.39–2.64)	0.035	2.31 (2.21–2.42)	2.57 (2.45–2.70)	0.001	2.52 (2.26–2.77)	2.36 (2.03–2.69)	0.245
NEAP (mEq/day)	50.5 (47.9–53.2)	46.1 (42.7–49.5)	0.017	54.1 (51.1–57.1)	46.7 (43.2–50.1)	<0.001	42.2 (35.1–49.2)	43.2 (34.0–52.5)	0.770-
Model 3				-	-				
Dietary fiber (g/day)	2.40 (2.35–2.45)	2.50 (2.41–2.59)	0.070	2.35 (2.26–2.44)	2.52 (2.41–2.62)	0.009	2.44 (2.36–2.52)	2.39 (2.21–2.58)	0.626
NEAP (mEq/day)	49.8 (48.1–51.5)	44.9 (42.0–47.9)	0.005	54.6 (51.7–57.4)	45.8 (42.5–49.2)	<0.001	46.9 (44.2–49.7)	47.5 (41.3–53.7)	0.867
Model 4				-	-	*-*	-	-	*-*
Log dietary fiber	2.38 (2.30–2.46)	2.47 (2.37–2.57)	0.088	2.35 (2.26–2.44)	2.52 (2.41–2.62)	0.010	2.48 (2.26–2.69)	2.38 (2.11–2.66)	0.407
NEAP (mEq/day)	50.6 (48.0–53.3)	45.7 (42.4–49.0)	0.007	54.6 (51.7–57.4)	45.8 (42.5–49.2)	<0.001	41.7 (34.8–48.7)	43.5 (34.4–52.5)	0.634-

Values for outcome variables are geometric means and 95% CI. NEAP, net endogenous acid production score. Model 1 is adjusted for age and sex. Model 2 is adjusted for Model 1 + duration of diabetes, the presence of hypertension, smoking, alcohol consumption, exercise, HbA1c, triglycerides and body mass index. Model 3 is adjusted for Model 1 + energy intake (kcal/ideal body weight/day). Model 4 is adjusted for Model 2 + energy intake (kcal/ideal body weight/day).

## Data Availability

The datasets generated during and/or analyzed during the current study are available from the corresponding author on reasonable request.

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
