# Peer review of "Household Income Is Related to Dietary Fiber Intake and Dietary Acid Load in People with Type 2 Diabetes: A Cross-Sectional Study"

_nutrients, 2022, doi:10.3390/nu14153229_

Round 1

Reviewer 1 Report

Comments on Nutrients-1823742 v1 “Household Income is Related to Dietary Fiber Intake and Dietary Acid Load in People with Type 2 Diabetes: A Cross-Sectional Study

In this cross-sectional study, 201 subjects with T2DM were involved in the analysis of association between household income and dietary fiber intake/acid load. The results from this study showed that there was an association between household income and dietary fiber intake/acid load in men but not in women. Here are reviewer’s comments for authors’ consideration.

1.       As the authors indicated that the participants of this cross-sectional study were recruited from a cohort study, it is necessary to brief the cohort to help readers understand the background of this analysis. For example, who were those participants? Why were they involved in the cohort study?

2.       As the household income was the main risk factor of interest in this analysis and participants involved were older adults, it would be grateful to have other characteristics being added into the table 1 if possible, such as married status, education, comorbidity, etc. those may give readers a comprehensive picture of the participants in this study.

3.       338 patients were involved at the beginning of the study, but 137 of them were excluded from the analysis. 24 of those being excluded were non-T2DM, for the rest of them who were T2DM being excluded, could an attrition analysis be conducted to examine whether there was any difference compared to those who have been included in the analysis in regards of their demographic characteristics, such as the proportion of gender or mean age. This may help evaluation of the generalizability of the study.

4.       Household income 5,000,000JPY was used as the cut-off in defining low or high household income. Was the criterium used in this study a recommended cut-off from the citation reference 22? If yes, please specify it.

5.       It seemed all data analyses were performed gender separately, which should be addressed in the methods to avoid a confusion. However, the reviewer has different opinions as present in point 8 below.

6.       Multivariable linear regression was used to calculate the adjusted means of dietary fiber intake and dietary acid load. two important assumptions of using the MLR are the dependent variable should be normal distributed and equal variance.  Not sure if authors have verified these assumptions, particularly, the PRAL had much larger SDs. Please add those results either in method or in results report.

7.       The factors adjusted for the means of dietary fiber intake and dietary acid load listed in table included age, BMI, and eight other variables. it is unclear how these variables were determined to be included into the model. carefully selecting covariates to be adjusted in the model is important.

8.       Analytic strategies recommended for dealing with the gender’s issue: a) table 1 & 2 (they can combined as one table) did show statistical significance for many variables between genders, but most of them showed a similar direction between low vs high income for both genders, eg, diabetes duration was 19.4 vs. 13.4 for men and 18.9 vs 14.3 for women; presence of hypertension 75.6% vs. 52.4% for men and 72.1% vs. 50.0% for women (BTW, no need to present both proportions (-/+) in the table, one is enough), therefore, it would be better to have gender combined analysis for table 3 - 5 (the current tables 3 & 4 can be supplementary tables). b) For table 3 & 4 (in fact, I would suggest pooling them together as one table as well), you can consider presenting the adjusted means or proportions with adjusting for age and gender. C) For table 5, four modeling strategies for authors’ consideration. first, income category variable, age and gender should be in all four models. Then consider adding other covariates in the four models separately. model 1, add those potential clinic confounding factors identified only; model 2, those habitual dietary intake variables identified only; model 3, add both variables from model 1 & 2; and model 4, in model 3 further add an interaction term of gender and income.  

Author Response

Response to Reviewer 1

Comments on Nutrients-1823742 v1 “Household Income is Related to Dietary Fiber Intake and Dietary Acid Load in People with Type 2 Diabetes: A Cross-Sectional Study”

In this cross-sectional study, 201 subjects with T2DM were involved in the analysis of association between household income and dietary fiber intake/acid load. The results from this study showed that there was an association between household income and dietary fiber intake/acid load in men but not in women. Here are reviewer’s comments for authors’ consideration.

Point 1

As the authors indicated that the participants of this cross-sectional study were recruited from a cohort study, it is necessary to brief the cohort to help readers understand the background of this analysis. For example, who were those participants? Why were they involved in the cohort study?

Response

Thank you for your comment. This cohort study, KAMOGAWA-DM cohort study, involved outpatients from the Department of Endocrinology and Metabolism, Kyoto Prefectural University of Medicine Hospital (Kyoto, Japan). The KAMOGAWA-DM cohort study has been running since 2014, and the goal of this cohort study is to reveal the natural history of people with diabetes. The patients were invited to participate by their primary doctors, and those who agreed were included in this cohort study. Therefore, we have added these points in the Methods section described as below.

Methods (Line: 65-70)

“This cross-sectional study was included in the prospective KAMOGAWA-DM cohort study, running since 2014 [19]. This cohort study involved outpatients from the Department of Endocrinology and Metabolism, Kyoto Prefectural University of Medi-cine Hospital (Kyoto, Japan). The goal of this cohort study is to reveal the natural history of people with diabetes. The patients were invited to participate by their primary doctors, and those who agreed were included in this cohort study. All participants provided written informed consent.”

Point 2

As the household income was the main risk factor of interest in this analysis and participants involved were older adults, it would be grateful to have other characteristics being added into the table 1 if possible, such as married status, education, comorbidity, etc. those may give readers a comprehensive picture of the participants in this study.

Response

Thank you for your valuable comment. We have added the presence of renal failure, educational level, and married status. Therefore, we have revised in Table 1, new Table 2 and the Methods section, described as below.

Methods (Line: 81-86)

“Additionally, participants were categorized as non-exercisers and exercisers based on their performance, or lack thereof, of any type of sport at least one time per week. Educational level was evaluated with the following response options: “elementary school,” “junior high school,” “high school,” “technical college,” “vocational school,” “college,” and “graduate school,” and educational background of “elementary school,” or “junior high school” was defined as <12 years [21].”

Methods (Line: 101)

“Renal failure was defined as eGFR <30 mL/min per 1.73 m2 [26].”

References

  1. Sakurai, M.; Nakagawa, H.; Kadota, A.; Yoshita, K.; Nakamura, Y.; Okuda, N.; Nishi, N.; Miyamoto, Y.; Arima, H.; Ohkubo, T.; et al. Macronutrient Intake and Socioeconomic Status: NIPPON DATA2010. J. Epidemiol. 2018, 28, S17–S22, doi:10.2188/jea.JE20170250.
  2. Haneda, M.; Utsunomiya, K.; Koya, D.; Babazono, T.; Moriya, T.; Makino, H.; Kimura, K.; Suzuki, Y.; Wada, T.; Ogawa, S.; et al. A New Classification of Diabetic Nephropathy 2014: A Report from Joint Committee on Diabetic Nephropathy. J. Diabetes Investig. 2015, 6, 242–246, doi:10.1111/jdi.12319.

Point 3

338 patients were involved at the beginning of the study, but 137 of them were excluded from the analysis. 24 of those being excluded were non-T2DM, for the rest of them who were T2DM being excluded, could an attrition analysis be conducted to examine whether there was any difference compared to those who have been included in the analysis in regards of their demographic characteristics, such as the proportion of gender or mean age. This may help evaluation of the generalizability of the study.

Response

Thank you for your comment. We have assessed the difference between included and excluded participants. HbA1c and exercise habit were different between included and excluded participants with T2DM. The difference of characteristics between included and excluded participants was not different, although HbA1c and exercise habit were different between included and excluded participants. We have added Supplemental Table 1.

Point 4

Household income 5,000,000JPY was used as the cut-off in defining low or high household income. Was the criterium used in this study a recommended cut-off from the citation reference 22? If yes, please specify it.

Response

Thank you for your comment. The average salary in Japan is reported to be 4,360,000 JPY, thus, in this study, the cut-off in defining low or high household income was set at 5,000,000 JPY, which was the closest among the options in this study. According to your comment, we have described this point in the Methods section as below.

Methods (Line: 89-92)

“The average salary at that time of this study was 4,360,000 JPY [23]. Therefore, household income of “<3,000,000 JPY” or “3,000,000–5,000,000 JPY” was defined as low household income, whereas that of “5,000,000–8,000,000 JPY” or “≥8,000,000 JPY” was defined as high household income in this study [23].”

Point 5

It seemed all data analyses were performed gender separately, which should be addressed in the methods to avoid a confusion. However, the reviewer has different opinions as present in point 8 below.

Response

Thank you for your comment. About these points, we have revised in the Methods section and new Table 2 and 3.

Methods (Line: 127-128)

“Moreover, because the characteristics and dietary intakes differed between men and women, the data was analyzed by sex.”

Point 6

Multivariable linear regression was used to calculate the adjusted means of dietary fiber intake and dietary acid load. two important assumptions of using the MLR are the dependent variable should be normal distributed and equal variance.  Not sure if authors have verified these assumptions, particularly, the PRAL had much larger SDs. Please add those results either in method or in results report.

Response

Thank you for your comment. As you say, the PRAL had much large SDs. To check the normal distribution of PRAL, NEAP and dietary fiber intake, Shapiro-Wilk test has been performed. We have found that NEAP is normal distributed, whereas PRAL and dietary fiber intake were not. After performing logarithmic transformation, log (dietary fiber intake) was normal distributed. Therefore, NEAP and log (dietary fiber intake) were used for multivariable linear regression. We have revised in the Methods and Results sections and new Table 3 as below.

Methods (Line: 129-136)

“NEAP was equal variance. Although dietary fiber intake was not equal variance, logarithmic dietary fiber intake was equal variance. Therefore, NEAP and log (dietary fiber intake) were used for multivariable linear regression to assess the association between household income and log (dietary fiber intake) and dietary acid load.  Multivariable linear regression analyses were executed, and geometric means with 95% confidence intervals were calculated, after adjusting for age, sex, BMI, the duration of diabetes, exercise habit, smoking habit, HbA1c, triglycerides, presence of hypertension, energy intake, and alcohol consumption.”

Results (Line: 181-188)

“Log (dietary fiber intake) with low household intake tended to be lower than that with high household income (2.38 [2.30-2.46] vs. 2.47 [2.37-2.57], p = 0.088). Log (dietary fiber intake) in men with low household income was lower than that in those with high household income after adjusting for covariates (2.35 [2.26-2.44] vs. 2.52 [2.41-2.62], p = 0.010). Furthermore, NEAP (54.6 [51.7-57.4] vs. 45.8 [42.5-49.2], p <0.001) in men with low household income were higher than in those with high household income after adjusting for covariates. In contrast, household income was not related to dietary fiber intake and dietary acid load in women after adjusting for covariates.”

Point 7

The factors adjusted for the means of dietary fiber intake and dietary acid load listed in table included age, BMI, and eight other variables. it is unclear how these variables were determined to be included into the model. carefully selecting covariates to be adjusted in the model is important.

Response

Thank you for your comment. Age, duration of diabetes, BMI, HbA1c, triglycerides and presence of hypertension are known to affect diet. Exercise, smoking and drinking alcohol affect glycemic control, which are associated with diet therapy, including dietary fiber intake. Increased energy intake results in a relatively high dietary fiber intake. Therefore, we have mentioned this point in the Methods section as below.

Methods (Line: 136-139)

“Age, duration of diabetes, BMI, HbA1c, triglycerides and presence of hypertension are known to effect diet [32-35]. Exercise, smoking and drinking alcohol affected glycemic control, which are associated with diet therapy, including dietary fiber intake [36-38]. Increased energy intake results in a relatively high dietary fiber intake.”

References

  1. Roberts, S.B.; Rosenberg, I. Nutrition and Aging: Changes in the Regulation of Energy Metabolism with Aging. Physiol. Rev. 2006, 86, 651–667, doi:10.1152/physrev.00019.2005.
  2. Mottalib, A.; Salsberg, V.; Mohd-Yusof, B.N.; Mohamed, W.; Carolan, P.; Pober, D.M.; Mitri, J.; Hamdy, O. Effects of Nutrition Therapy on HbA1c and Cardiovascular Disease Risk Factors in Overweight and Obese Patients with Type 2 Diabetes. Nutr. J. 2018, 17, 1–10, doi:10.1186/s12937-018-0351-0.
  3. Preuss, H.G.; Gondal, J.A.; Lieberman, S. Association of Macronutrients and Energy Intake with Hypertension. http://dx.doi.org/10.1080/07315724.1996.10718561 2013, 15, 21–35, doi:10.1080/07315724.1996.10718561.
  4. Riccardi, G.; Rivellese, A.A. Effects of Dietary Fiber and Carbohydrate on Glucose and Lipoprotein Metabolism in Diabetic Patients. Diabetes Care 1991, 14, 1115–1125, doi:10.2337/DIACARE.14.12.1115.
  5. Tsukui, S.; Kanda, T.; Nara, M.; Nishino, M.; Kondo, T.; Kobayashi, I. Moderate-Intensity Regular Exercise Decreases Serum Tumor Necrosis Factor-α and HbA1c Levels in Healthy Women. Int. J. Obes. 2000 249 2000, 24, 1207–1211, doi:10.1038/sj.ijo.0801373.
  6. Nilsson, P.M.; Gudbjörnsdottir, S.; Eliasson, B.; Cederholm, J. Smoking Is Associated with Increased HbA1c Values and Microalbuminuria in Patients with Diabetes — Data from the National Diabetes Register in Sweden. Diabetes Metab. 2004, 30, 261–268, doi:10.1016/S1262-3636(07)70117-9.
  7. Hirakawa, M.; Arase, Y.; Amakawa, K.; Ohmoto-Sekine, Y.; Ishihara, M.; Shiba, M.; Ogawa, K.; Okuda, C.; Jinno, T.; Kato, H.; et al. Relationship between Alcohol Intake and Risk Factors for Metabolic Syndrome in Men. Intern. Med. 2015, 54, 2139–2145, doi:10.2169/INTERNALMEDICINE.54.2736.

Point 8

Analytic strategies recommended for dealing with the gender’s issue: a) table 1 & 2 (they can combined as one table) did show statistical significance for many variables between genders, but most of them showed a similar direction between low vs high income for both genders, eg, diabetes duration was 19.4 vs. 13.4 for men and 18.9 vs 14.3 for women; presence of hypertension 75.6% vs. 52.4% for men and 72.1% vs. 50.0% for women (BTW, no need to present both proportions (-/+) in the table, one is enough), therefore, it would be better to have gender combined analysis for table 3 - 5 (the current tables 3 & 4 can be supplementary tables). b) For table 3 & 4 (in fact, I would suggest pooling them together as one table as well), you can consider presenting the adjusted means or proportions with adjusting for age and gender. C) For table 5, four modeling strategies for authors’ consideration. first, income category variable, age and gender should be in all four models. Then consider adding other covariates in the four models separately. model 1, add those potential clinic confounding factors identified only; model 2, those habitual dietary intake variables identified only; model 3, add both variables from model 1 & 2; and model 4, in model 3 further add an interaction term of gender and income.  

Response

Thank you for your valuable comment. According to your comment, we have combined Table 1 and 2, and added gender combined analysis for new Table 2 and 3. However, the percentage of men between low and high household income was significantly different, we have remained gender separated analysis. We have revised new Table 3. Model 1 is adjusted for age and sex; Model 2 is adjusted for age, sex, duration of diabetes, BMI, the presence of hypertension, energy intake, exercise, smoking, alcohol consumption, triglycerides, and HbA1c; Model 3 is adjusted for age, sex, energy intake, and Model 4 is adjusted for Model 2 and energy intake. Therefore, we have revised in the Methods and Results sections and new Table 3.

Methods (Line: 136-139)

“Age, duration of diabetes, BMI, HbA1c, triglycerides and presence of hypertension are known to effect diet [32-35]. Exercise, smoking and drinking alcohol affected glycemic control, which are associated with diet therapy, including dietary fiber intake [36-38]. Increased energy intake results in a relatively high dietary fiber intake.”

Results (Line: 165-169)

“People with low household intake was older than those with high household intake (70.4±7.7 vs. 65.3±10.4 years, p <0.001). The percentage of men in people with low household intake was lower than that with high household intake (58.5 vs. 77.8 %, p = 0.019). Dietary fiber intake in people with low household income was lower than that in those with high household income (11.7 ± 4.5 vs. 13.5 ± 5.9 g/day, p = 0.028).”

Results (Line: 181-188)

“Log (dietary fiber intake) with low household intake tended to be lower than that with high household income (2.38 [2.30-2.46] vs. 2.47 [2.37-2.57], p = 0.088). Log (dietary fiber intake) in men with low household income was lower than that in those with high household income after adjusting for covariates (2.35 [2.26-2.44] vs. 2.52 [2.41-2.62], p = 0.010). Furthermore, NEAP (54.6 [51.7-57.4] vs. 45.8 [42.5-49.2], p <0.001) in men with low household income were higher than in those with high household income after adjusting for covariates. In contrast, household income was not related to dietary fiber intake and dietary acid load in women after adjusting for covariates.”

References

  1. Roberts, S.B.; Rosenberg, I. Nutrition and Aging: Changes in the Regulation of Energy Metabolism with Aging. Physiol. Rev. 2006, 86, 651–667, doi:10.1152/physrev.00019.2005.
  2. Mottalib, A.; Salsberg, V.; Mohd-Yusof, B.N.; Mohamed, W.; Carolan, P.; Pober, D.M.; Mitri, J.; Hamdy, O. Effects of Nutrition Therapy on HbA1c and Cardiovascular Disease Risk Factors in Overweight and Obese Patients with Type 2 Diabetes. Nutr. J. 2018, 17, 1–10, doi:10.1186/s12937-018-0351-0.
  3. Preuss, H.G.; Gondal, J.A.; Lieberman, S. Association of Macronutrients and Energy Intake with Hypertension. http://dx.doi.org/10.1080/07315724.1996.10718561 2013, 15, 21–35, doi:10.1080/07315724.1996.10718561.
  4. Riccardi, G.; Rivellese, A.A. Effects of Dietary Fiber and Carbohydrate on Glucose and Lipoprotein Metabolism in Diabetic Patients. Diabetes Care 1991, 14, 1115–1125, doi:10.2337/DIACARE.14.12.1115.
  5. Tsukui, S.; Kanda, T.; Nara, M.; Nishino, M.; Kondo, T.; Kobayashi, I. Moderate-Intensity Regular Exercise Decreases Serum Tumor Necrosis Factor-α and HbA1c Levels in Healthy Women. Int. J. Obes. 2000 249 2000, 24, 1207–1211, doi:10.1038/sj.ijo.0801373.
  6. Nilsson, P.M.; Gudbjörnsdottir, S.; Eliasson, B.; Cederholm, J. Smoking Is Associated with Increased HbA1c Values and Microalbuminuria in Patients with Diabetes — Data from the National Diabetes Register in Sweden. Diabetes Metab. 2004, 30, 261–268, doi:10.1016/S1262-3636(07)70117-9.
  7. Hirakawa, M.; Arase, Y.; Amakawa, K.; Ohmoto-Sekine, Y.; Ishihara, M.; Shiba, M.; Ogawa, K.; Okuda, C.; Jinno, T.; Kato, H.; et al. Relationship between Alcohol Intake and Risk Factors for Metabolic Syndrome in Men. Intern. Med. 2015, 54, 2139–2145, doi:10.2169/INTERNALMEDICINE.54.2736.

Reviewer 2 Report

In my opinion manuscript is quite well prepared. Before possible publication, please correct or supplement:

1) please add the source/reference to the used BDHQ questionnaire - please give limitations for this questionnaire,

2) please correct the data in figure 1 (it should be 128 men and 73 women as in the other tables),

3) lines 8-9 from the bottom on page 7 of the work - please complete the sentence in the discussion (the beginning of the sentence "Previously, a relationship between household income ..."),

4) why were 4 income thresholds given if only two were analyzed? In my opinion, it is worth presenting the results for extreme incomes (below 3,000,000 JPY and above 8,000,000 JPY).

Author Response

Response to Reviewer 2

In my opinion manuscript is quite well prepared. Before possible publication, please correct or supplement:

Point 1

please add the source/reference to the used BDHQ questionnaire - please give limitations for this questionnaire,

Response

Thank you for your valuable comment. We have added the reference on the validity of the nutrient intake in BDHQ. A previous study showed that the data were unreliable for those with <600 or more >4000 kcal/day, therefore, they were excluded. There was another limitation that the Pearson correlation coefficients between the dietary record and the BDHQ was around r = 0.60. We have mentioned this point in the Methods and Discussion sections as below.

Methods (Line: 75-77)

“The exclusion criteria were non-T2DM, extremely low or high energy intake (<600 or >4000 kcal/day), as extremely low or high energy intake is unnatural [20], incomplete questionnaire; and unknown household income.”

Methods (Line: 109-110)

“The details and validity of BDHQ have been presented previously [27].”

Discussion (Line: 241-244)

“Fourth, the validation of BDHQ has been showed previously [27]. However, the Pearson correlation coefficients between the dietary record and the BDHQ is around r = 0.60, which is a little low.”

References

  1. Kobayashi, S.; Honda, S.; Murakami, K.; Sasaki, S.; Okubo, H.; Hirota, N.; Notsu, A.; Fukui, M.; Date, C. Both Comprehensive and Brief Self-Administered Diet History Questionnaires Satisfactorily Rank Nutrient Intakes in Japanese Adults. J. Epidemiol. 2012, 22, 151–159, doi:10.2188/jea.JE20110075.

Point 2

please correct the data in figure 1 (it should be 128 men and 73 women as in the other tables),

Response

Thank you for your valuable comment. As your comment, we have revised it.

Point 3

lines 8-9 from the bottom on page 7 of the work - please complete the sentence in the discussion (the beginning of the sentence "Previously, a relationship between household income ..."),

Response

Thank you for your comment. We have revised it as below.

Discussion (Line: 223-224)

“Previously, a relationship between household income and glycemic control in people with T2DM has been found [5].”

Point 4

why were 4 income thresholds given if only two were analyzed? In my opinion, it is worth presenting the results for extreme incomes (below 3,000,000 JPY and above 8,000,000 JPY).

Response

Thank you for your comment. A previous study used 4 income thresholds, and this study also used those. However, the statistical participants were small. Therefore, we have the cut-off in defining low or high household income based on the average salary at that time of this study. As your say, presenting the results for extreme incomes is very important. Unfortunately, the participants, especially below 3,000,000 JPY and above 8,000,000 JPY, were not enough. We think it is good to consider increasing the number of participants to be analyzed in the future. We have added this point as one of the limitations of this study in the Discussion section described as below.

Discussion (Line: 238-240)

“Moreover, the number of participants, especially extreme incomes, were not enough. Therefore, we need further research with more participants and used the different cut-off.”